# Effect of Surface Roughness on Pitting Corrosion of 2205 Duplex Stainless Steel Investigated by Electrochemical Noise Measurements

**DOI:** 10.3390/ma12050738

**Published:** 2019-03-04

**Authors:** Yiwei Tang, Nianwei Dai, Jun Wu, Yiming Jiang, Jin Li

**Affiliations:** Department of Materials Science, Fudan University, Shanghai 200433, China; tangyiwei1991@gmail.com (Y.T.); 16110300022@fudan.edu.cn (N.D.); 16110300024@fudan.edu.cn (J.W.); ymjiang@fudan.edu.cn (Y.J.)

**Keywords:** duplex stainless steel, electrochemical noise, pitting corrosion, surface roughness

## Abstract

The influence of surface roughness on the pitting corrosion behaviour of 2205 duplex stainless steel (DSS) in a chloride-containing environment was investigated using electrochemical noise (EN) techniques and morphology observation. A rougher surface condition increased the frequency of pit initiation because of the increase in more occluded pit sites. Rough surface finish also accelerated pit growth by increasing the actual dissolution rate in the pit. Metastable pits on rougher surfaces had longer lifetimes and grew to larger sizes, as their inner chemical environment was more easily maintained. However back-scatter images showed that pitting initiates on DSS 2205 regardless of the roughness condition.

## 1. Introduction

Duplex stainless steel (DSS) is the dual-phase structure stainless steel that contains almost equal contents of ferrite and austenite and has been widely applied in the marine and petroleum industries [1,2]. Since DSS is used in chloride-containing environments, the study of its pitting corrosion resistance of is often the focus of research [3,4]. According to previous research, many factors have an effect on pitting corrosion, and surface roughness is one that cannot be neglected. The influence of surface roughness on metastable pitting, critical pitting temperature (CPT), and pitting potential of austenite stainless steel has been studied [5,6]. An increase in surface smoothness leads to an increase in CPT of austenite stainless steel. Meanwhile, a smoother surface finish reduces the incidence of metastable pitting and increases the pitting potential of austenite stainless steel in chloride solution, but it is doubted whether the conclusion can be fully extended to DSS. Besides, the specific effect of surface roughness on pitting initiation and growth stage on stainless steel is seldom reported.

To investigate further the pitting corrosion of stainless steel, especially the nucleation process, electrochemical noise (EN) was introduced into the research. Electrochemical noise techniques were first applied to corroding metals by Iverson in 1968 [7], and it has been diversified and widely utilized in recent decades because of the improvement in electronic devices and signal analytical methods. Compared to other electrochemical methods, EN measurement has the advantage of recoding the early stage of localized corrosion initiation in real time with high sensitivity [8,9]. In addition, in contrast to conventional electrochemical techniques, EN measurement does not require excitation signals. Because of these advantages, EN signals are widely accepted in the monitoring of localized corrosion of metallic materials [10,11,12]. With the development of these techniques, EN signal analysis has been used as a measurement of the severity of localized corrosion in DSS [13]. Recently, the application of EN measurement in investigating the initiation and passivity processes of localized corrosion of stainless steels under different polarization has also been reported. Kim [14] performed EN wavelet analysis in detecting the stable and metastable pitting in NaCl solution, and found great applicability for this method. Klapper et al. [15] observed that a strong cathodic process can change the form and reduce the amplitude of EN signals arising from pitting corrosion in stainless steel. Thus, EN technique can be a more suitable method for investigating the influence of the roughness factor on the initiation and growth of pitting.

In the present work, the object was to study the influence of surface roughness on the initiation and early stage of the growth process of pitting corrosion in DSS. Electrochemical noise measurement was carried out on 2205 DSS with different surface roughness conditions both at open circuit potential (OCP) and under anodic polarization.

## 2. Materials and Methods

### 2.1. Material and Specimen Preparation

The alloy used in this work was 2205 duplex stainless steel (DSS 2205, Baosteel, Shanghai, China); the chemical composition of the material is listed in Table 1.

All the specimens were solution-annealed at 1050 °C for 1 h followed by water quenching to obtain better initial pitting resistance [16,17]. The microstructure of an annealed specimen is shown in Figure 1. The non-working surface of the specimen was soldered to the copper wire for electrical connection. Then the specimens were sealed in epoxy resin, exposing a 10 mm × 10 mm working surface for the electrochemical tests. Finally, the working surface was ground with abrasive paper to four alternative finishes: 180#, 600#, 1000#, and 2000#. An extra specimen, polished to 2.5 μm with diamond polishing paste, and was included in the test. Before the experiments, all specimens were degreased with ethanol, rinsed with distilled water, and dried in an oven for at least 4 h.

### 2.2. EN Experiments and Analysis

The EN measurements were conducted using a PARSTAT™ MC multi-channel potentiostat/galvanostat with an integrated zero-resistance galvanometer (ZRA, Princeton Applied Research, Oak Ridge, TN, USA). The reference electrode used in all electrochemical tests was a saturated calomel electrode (SCE, INESA, Shanghai, China).

For the EN measurement at OCP, two specimens of the same surface roughness were used as working electrodes. The electrolyte was a solution of 10% FeCl_3_·6H_2_O (Sinopharm, Shanghai, China), and throughout the whole test the solution temperature was kept at 50 °C [13]. Immediately after the 1 h immersion of the specimens, the EN data were recorded for 3600 s and separated into 10 sections: each section of 360 s containing 3686 data points, so the sampling rate was about 10 Hz. The direct potential and current offset of the EN data were removed by using a 5-order polynomial detrending method [18].

In the potentiostatic EN measurements, a platinum electrode was used as counter electrode. The test electrolyte was 1 M NaCl solution. The applied potential and the experimental temperature were kept at 200 mV_SCE_ and 50 °C, respectively. The EN data of each sample were recorded for 1200 s at a sampling rate of 20 Hz and a 5-order polynomial used to remove the direct current offset. The pitting event number was calculated according to the records of current transient peaks. The effective current transient of metastable pitting was identified as the current peak at least 100 nA higher than the background current noise.

### 2.3. Surface Observation and Characterization

The microstructure of solution-annealed specimens was characterized through scanning electron microscopy (SEM, JSM-6701F, JEOL, Tokyo, Japan).

The surface morphology after EN measurement was observed by optical metallographic microscope. Every specimen was tested at least three times to ensure reproducibility of the experiments.

The morphological characterization of the specimens after potentiostatic measurement was performed by a back-scattered electron (BSE) technique using SEM (XL30FEG, Philips, Eindhoven, The Netherlands).

## 3. Results and Discussion

### 3.1. Electrochemical Noise Analysis at OCP

The typical EN patterns of DSS 2205 samples of different surface roughness in 10% FeCl_3_ at 50 °C are shown in Figure 2. The time domain spectra of the EN data show that the current noise fluctuation of different surface roughness samples stayed at the same level over the entire measurement. These high-frequency fluctuations in current arose from the instability of the passive film on all specimens. However, the noise potential was shifted in the positive direction, from 100 to 500 mV_SCE_, with the increase in surface roughness, and the fluctuation of the potential noise also increased. Considering the cross-correlation between current noise and potential noise, this phenomenon indicates that the electrochemical impedance decreased with the increase in surface roughness. Thus, a rougher sample exhibited a more active surface state as well as a higher possibility of stable pitting. This EN measurement results correspond to the optical microscopy images of the specimens after the test, which are displayed in Figure 3.

Based on the analysis of these time domain spectra after removing both the direct potential and current drift, the EN resistance (*R_n_*) can be calculated according to followed equation:
(1)$$R_{n}=\frac{\sigma_{E}}{\sigma_{I}}$$
where *σ_E_* is the standard deviation (SD) of the noise potential and *σ_I_* is the SD of noise current. *R_n_* is thought to be equivalent to the polarization resistance *R_p_* [19], assuming that the difference in the effective frequency at which *R_n_* and *R_p_* are measured is negligible [20]. The frequency used in this measurement was not close to that used in normal *R_p_* measurement, but *R_p_* can still be estimated with a reasonable error [21]; therefore, *R_n_* is comparable to *R_p_* and reflects the corrosion resistance on the material surface. The reciprocals of calculated *R_n_* of different surface roughness specimens are presented in Figure 4. The 1/*R_n_* data characterized the change in corrosion rate on the sample surface during the measuring time. It was observed that the specimen with the polished surface presented the lowest 1/*R_n_* value, at 4.6 × 10^−5^ kΩ^−1^·cm^−2^, and the smallest amplitude fluctuation, suggesting that pit growth did not emerge frequently on the smooth surface. In contrast, the 180# specimen showed a higher corrosion rate at the beginning of the measurement, which declined quickly, indicating a combination of pit initiation and repassivation. Moreover, in the latter half of the recording, the 1/*R_n_* value of the 180# specimen rose to 1500 kΩ^−1^·cm^−2^ and remained at the same level, which is indicative of stable pit growth. The corrosion rate of 600#, 1000#, and 2000# specimens reveal a trend similar to that of the 180# specimen, but slower as the surface became smoother. The variation of 1/*R_n_* implies that lower surface roughness has a positive effect in enhancing the pit corrosion resistance of DSS by reducing the probability of stable pit growth. This is similar to the trend of austenite stainless steel, indicating that mechanical grinding has the same effect on the pitting corrosion resistance of DSS, although it has the particular dual-phase structure.

For more information on localized corrosion, the frequency domain analysis of potential and current signal was conducted. The power spectral density (PSD) of potential and current noise during the whole immersion time (3600 s) was calculated by fast Fourier transform (FFT) with “Hanning” window, as shown in Figure 5 and Figure 6, respectively. It can be seen in Figure 5 that current PSDs among different surface roughness specimens present similar values and trends, which are consistent with the results obtained from the preceding time domain analysis. In addition, the cut-off frequencies of the current PSDs are around 2 × 10^−2^ Hz, and a low-frequency plateau of white noise is evident in the plots.

In contrast, the PSDs of potential noise for different surface roughness specimens show much greater differences, as illustrated in Figure 6. At a frequency lower than 10^−3^ Hz, the potential PSDs also show the plateau. At higher frequency, the potential PSDs of rougher surfaces (180#, 600#) are evidently higher than the smoother ones (1000#, 2000# and polished), suggesting that the number of transients on rougher surfaces increased [22] due to the geometrical factor of surface roughness. The geometries of surface pit sites, including the openness of the pit mouth and the diffusion length to bulk solution, are considered to govern the diffusion rate and consequently control the pit initiation and growth [5]. However, the PSD plots cannot clearly discriminate the different surface roughness specimens from each other. Therefore, further frequency domain analysis of potential and current EN was applied.

Considering the low-frequency plateaus in both potential and current PSD plots, there might be a relatively pure shot noise process resulting from electrochemical process on the material surface [23]. Thus, it is feasible to use shot noise analysis on the PSD results. Shot noise theory is based on the fact that current is carried by discrete charge carriers, and consequently the number of the charge carriers passing a given point will be a random variable [24]. It is assumed that process of film breakdown, pit initiation and metal dissolution can be treated as shot noise with a corrosion current *I_corr_* and a charge of *q* in each event. Accordingly, the average corrosion current $\bar{I_{corr}}$ is given by:
(2)$$\bar{I_{corr}}=\bar{q}\cdot f_{ns}$$
where $\bar{q}$ is the average charge in each event and *f_ns_* is the frequency of the event over the specimen. Thus, based on the shot noise formula and the Stern–Geary equation [21], the frequency of the event per unit area *f_n_* is given by:(3)$$f_{n}=\frac{\bar{I_{corr}}}{\bar{q}\cdot A}=\frac{B^{2}}{\Psi_{E}\cdot A}$$
where *B* is the Stern–Geary constant, *A* is the exposed area of the specimen and *Ψ**_E_* is the potential PSD at the low-frequency limit. In this work, the frequency of *Ψ**_E_* was chosen at around 10^−2^ Hz. Based on the results obtained from the common cathodic and anodic Tafel slope at OCP, a value of 0.168 V·s^−1^ for B was used in this analysis. This value was applied with the assumption that the relative behaviours of various systems will not affected by the variations of B [25].

On the grounds of current and potential PSDs of each recording period, the cumulative frequency of *f_n_* for different surface roughness specimens were calculated as *n*/(*N* + 1), where *n* is the number of the *f_n_* value in frequency counting, and *N* is the total number of *f_n_*. The results are plotted in Figure 7. Obviously, the *f_n_* values of all different surface roughness DSS2205 specimens are less than 1500 Hz·cm^−2^. The relatively small *f_n_* indicates the occurrence of pitting corrosion rather than uniform corrosion [26], which conforms to the morphology observations. Moreover, the average *f_n_* value and the distribution range of *f_n_* increase along with surface roughness. Polished-surface specimens produced the lowest *f_n_* (lower than 2 Hz·cm^−2^) and the narrowest distribution width of *f_n_* (from 0.05 to 2 Hz·cm^−2^) over the entire immersion duration, indicating pit initiation was suppressed on the smooth surface. In contrast, the 180# specimen exhibited the highest *f_n_* value, as well as the widest distribution range of *f_n_* (from 0.5 to 1000 Hz·cm^−2^). The increase in event frequency is because of the comparatively occluded primary cavity on a rough surface which can form a diffusion barrier and maintain the internal chemical environment. Besides, pit nucleation areas on rougher surfaces have a much greater diversity in shapes and sizes [27]. As a result, the variance in possible pit occurrence on rough-surface specimens is more significant. This may be the explanation for the broadening of the *f_n_* distribution range as the surface becomes rougher. For 600#, 1000# and 2000# specimens, the variation trends of *f_n_* were still the same; however, it is observed that the difference in *f_n_* values is poorly discriminated by the curve shape. Accordingly, *f_n_* effectively reflects that the increase of surface roughness increases the possibility of pitting nucleation on DSS, which is similar to that on austenite stainless steel, but the distinction is not clear when the difference in surface roughness is relatively small.

Furthermore, the charge in event *q* during each recording period was also calculated, which is expressed as follows [21]:(4)$$q=\frac{\sqrt{\Psi_{E}}\sqrt{\Psi_{I}}}{B}$$
where *Ψ**_E_* and *Ψ**_I_* are the potential and current PSD at around 10^−2^ Hz, *B* is. the Stern–Geary constant.

This parameter can estimate the mass loss in each metastable pitting pulse, and therefore reflect the pit growth process under different surface conditions. The cumulative frequency of *q* is presented in Figure 8. Clearly, the average value of *q* decreases in the order 180# > 600# > 1000# > 2000# > polished, and difference in *q* between the roughest and smoothest specimen is higher than one order of magnitude. Hence, a rougher surface provides a higher probability for a metastable pit to propagate larger in size and transition to a stable pit on DSS. Nevertheless, the *q* values for 600#, 1000# and 2000# samples still provide relatively poor distinctions between them. In summary, the short noise analysis parameter *q* can characterize the facilitation effect of a rough surface on the pit growth process of DSS 2205, which is similar to austenite stainless steel, but the discrimination ability needs to be improved by optimized statistical analysis.

### 3.2. Potentiostatic Electrochemical Current Noise Analysis

Since EN measurements at OCP could not provide sufficient distinction between different surface roughness samples, the EN measurement under potentiostatic control was conducted to investigate further the influence of surface roughness on metastable pitting. The potentiostatic electrochemical current noise (ECN) measurement in NaCl solution was first carried out at five different applied potentials (200, 300, 400, 500 and 600 mV_SCE_) and four solution temperatures (45, 50, 55 and 60 °C) to determine the optimal experimental parameters. Figure 9 shows the temperature and potential thresholds of stable pitting of DSS 2205 specimens of different surface roughness. For rougher-surfaced specimens, significant declines in pitting potential at all test temperatures are observed, especially in the temperature range 55–60 °C, which is close to the CPT of DSS 2205 [28,29]. This phenomenon indicates that the CPT of DSS 2205 specimens decrease with surface roughness. According to the new CPT model established by Li et al. [30], stable pitting occurs when the maximum pit dissolution current density is greater than or equal to the diffusion current density relative to the critical concentration maintained in an open pit. In this case, the initial cavity for pit nucleation was more occluded, and the internal electrochemical environment more easily maintained, on a rough surface. Hence the critical diffusion current density *i_diff_* decreases and consequently the transition from metastable to stable pitting takes place at lower applied potential and temperature.

Based on early potentiostatic test results, the applied potential for ECN measurement was chosen as 200 mV_SCE_, which is in the repassivation range of polarization for different surface roughness specimens at all four solution temperatures. Figure 10 shows the typical ECN time records of different surface roughness specimens at 55 °C. Apparently, the number and peak values of current transients that represent metastable pits were affected by surface roughness. Therefore, statistical analysis was performed on potentiostatic ECN data.

#### 3.2.1. Influence of Surface Roughness on Metastable Pit Nucleation

The metastable pitting rate *N* can be defined as the number of metastable pits per unit area per unit time and is used to characterize the pitting initiation rate of the material. The calculated results of DSS 2205 specimens of different roughness are presented in Figure 11. For all specimens, the current transients occur more frequently at higher temperature. A remarkable increase from 0.3 to 5.0 min^−1^·cm^−2^ of *N* on the 180# specimen was observed. Even for the polished specimen, there was a less obvious increase of 0.5 min^−1^·cm^−2^.

This phenomenon is consistent with the CPT model proposed by Li et al. [30]. The maximum pit dissolution current density *i_dis,max_*, which can describe charge-transfer-controlled dissolution, has a temperature dependence according to the Arrhenius equation:(5)$$i_{dis,max}=A\cdot \exp\left(\frac{-\Delta G}{RT}\right)$$
where ΔG is the activation energy of active metal dissolution, *A* represents the pre-exponential factor, *R* is the gas constant and *T* is the temperature. When temperature rises, the actual dissolution of metal in the pit site increases the con centration of metal cations inside the pit, approaching to critical concentration of metal cations *C_crit_*, which is the lowest value for the pit surface to maintain active dissolution [30]. As a result, the metastable pitting rate increases at higher temperature.

In addition, it was observed that rougher surface increased the frequency of metastable pitting increased at given temperature. Herein, the average *N* of the 180# specimen was more than seven times of that of the surface-polished specimen at all four testing temperatures. The values of *N* for other surface roughness specimens were intermediate and decreased with the successive increase in grit number. It is generally believed that a cruder surface morphology possesses a larger quantity of more occluded defective sites, in which it is easier to maintain an aggressive local environment [31]. As a result, the critical concentration *C_crit_* for pit nucleation is easier to be satisfied in these sites. By virtue of this, a rougher surface condition specimen has a higher probability of pitting nucleation, i.e., a higher metastable pitting rate in this work.

#### 3.2.2. Influence of Surface Roughness on Metastable Pit Growth Rate

According to the current transient time records shown in Figure 10, three different types of ECN transients were observed and their typical shapes are summarised in Figure 12.

Based on the plots, the ECN transient with a slow increase followed by a sharp decay (type II) was observed to be the major type of ECN transients detected, especially when the measured specimen was at a high metastable pitting rate. This shape feature indicates that the anodic amplitude of metastable pit growth was lower than that of the cathodic reaction, which means that the metal dissolution in the metastable pit was under charge-transfer control. Therefore, a type II ECN transient can represent metastable pit growth in this case, and the growth kinetics of the metastable pit can be described as *I*~*t^2^* on the basis of the shape and the rate-determining factors [32]. Taking all above conclusions into account, the slope of the rising part of type II ECN transient could characterize the pitting growth rate *κ* of a single initiated pit, which is defined as [33]:(6)$$\kappa =\frac{I_{n}-I_{1}}{t_{n}-t_{1}}$$
where *n* ≥ 1, *I* is the pitting current of a single pit, and *t* is the pit growth time. Considering the short duration of the ECN transient, the calculation of *κ* can be simplified by dividing the peak current value of the type II transient by the time taken to reach it [9]. Herein, the average metastable pitting growth rates (*Κ*) of different surface roughness DSS 2205 specimens at four temperatures and their SDs were calculated, and the results are illustrated in Figure 13. It is observed that metastable pit growth rate is clearly temperature dependent. The value of *K* at 60 °C is at least 20 times that at 45 °C for all roughness specimens. The increase of *K* can be ascribed to the positive effect of temperature on the electrochemical dissolution process of a metal, based on the Arrhenius equation (as shown in Equation (5)). In addition, the influence of temperature on ion transportation also contributes to the change of *K*. The rise in temperature accelerates the migration of chloride ions into the pit and consequently promotes metastable pit growth [34].

When considering the results obtained at same temperature, the value of *K* is observed to be higher on rougher-surface specimens. For instance, at 60 °C, the *K* value of the 180# specimen was 16 μA·cm^−2^, which is approximately 8–9 μA·cm^−2^ higher than that of 600# and 1000# specimens. As for the 2000# and 1000# specimens, a remarkable decrease in metastable pit growth rate was found: their values of *K* were in the range 0.5–2 μA·cm^−2^, only about 6% of that of the 180# specimen. When metastable pit maintain propagating, *i_dis,max_* should be equal or greater than limited diffusion current density *i_lim_*. That means that the actual dissolution rate of metal at the pit surface equals the actual diffusion rate of metal cations into bulk solution; the metastable pit growth is under diffusion control [35,36]. Because *i_lim_* is decided mainly by the geometry of the possible initiation sites [27], the value of *K* depends on the diffusion transport of metal cations associated with the surface morphology. As in this measurement, rougher surfaces provided deeper troughs in the surface profile; i.e., more occluded sites with shorter effective diffusion length *h* [27]. The reduction in *h* resulted in the decrease of *i_lim_*, which means that the transport of metal cations out of the pit interior was blocked and the local electrochemical environment was maintained sufficiently aggressive. The high concentration of metal cations and appropriate chloride counter anion would lead to a low pH inside the pit and consequently cause the rapid propagation rate of metastable pits [37]. The relationships between surface roughness, diffusion rate of metal cations in the pit and the dissolution rate of the metastable pit are shown schematically in Figure 14.

Furthermore, the SD of *K* was notably increased as specimen surface roughness increased. For the 180# specimen, the SD of *K* was highest, especially at higher solution temperature. As can be seen in Figure 13, the SD value of *K* for the 180# specimen at 60 °C reached 16 μA·cm^−2^, which is at least one order of magnitude higher than specimens with relatively smoother surfaces. The SD of the 600# and 1000# samples showed similar values and were clearly lower than for the 180# specimen. In comparison, the 2000# and polished samples exhibited the lowest SD levels. This is the result of the relatively wider distribution range of pit depth on rougher surfaces. The diversity of pit morphologies increases with the enhancement of surface roughness. Therefore, the local chemical environment of each individual pit site, which affects the growth rate of a metastable pit, is more distinctive on a coarser surface. This phenomenon correlates with the significant difference between corresponding ECN transients of rough surface samples in potentiostatic measurement.

#### 3.2.3. Influence of Surface Roughness on Metastable Pit Size

From the ECN time records in Figure 10, the ECN transients are all observed to be isolated, so each transient could be considered as corresponding to a single metastable pit. With this restriction, the quantity of metal dissolving in one metastable pitting process is proportional to the charge quantity of the corresponding ECN transient, and can be calculated based on Faraday’s law [32,33], as given by:(7)$$m=\frac{M\cdot Q}{nF}=\frac{M}{nF}\int_{0}^{t}Idt$$
where *m* is the quantity of metal dissolving, *Q* represents the charge quantity, *M* the atomic mass (58.8 for DSS 2205), *n* the valence state of metal cations (with an average of 2.2), *F* the Faraday constant, *I* the current of the ECN transient and *t* the transient lifetime. According to hemispherical model [35], *m* can be acquired by:(8)$$m=\frac{2\pi r^{3}\rho }{3}$$
where *r* is the radius of the pit and *ρ* is the density of the stainless steel (7.93 g·cm^−3^). Therefore, the volume and radius of the metastable pit is proportional to the charge quantity. In other words, the charge quantity of transient obtained from ECN results represents the metastable pit growth geometry on the sample surface.

The calculated average charge quantity *q* of the ECN transient is shown in Figure 15. As can be seen, at temperatures below 55 °C, the 180# specimen and the polished specimen exhibit the largest and smallest charge quantity, respectively. However, no remarkable difference between *q* values of 600#, 1000# and 2000# specimens is observed; when temperature increases to 60 °C, the difference in *q* between 600#, 1000# and 2000# specimens also increases. These observations indicate that metastable pit size increases on surfaces with greater roughness. Additionally, the highest SD of *q* is again observed on the 180# grit specimen at all test temperatures: from approximately 1 μC at 45 °C to 6 μC at 60 °C. In contrast, the 2000# and polished specimens show marginal SD values of *q*. Accordingly, the SD of charge quantity also showed an increasing trend as surface roughness increased.

In order to understand more directly the influence of surface roughness on metastable pit size, the average radius of metastable pits on specimens at different temperatures was calculated, as shown in Figure 16. A similar varied trend in metastable pit size can be observed from the calculated radius results. It is also observed that the radius of metastable pit is in the range of 1 to 10 μm, indicating that a metastable pit with greater radius may transition to a stable one; on the other hand, the shorter-radius metastable pits are hard to distinguish from the background current noise. The SD of the metastable pit radius still increased on rougher surfaces as well, but the values were all of the same order of magnitude.

These phenomena are ascribed to morphology of pit sites associated with surface roughness. The deeper and narrower cavity sites on rougher surface have longer effective diffusion lengths and hence hinder the diffusion of metal cations to the bulk solution [38]. This diffusion barrier maintains a sufficiently aggressive local environment inside the pit and promotes metal dissolution. Therefore *i_diff_*, which is associated with maintenance of the critical local environment (*C_cirt_*), decreases on rougher surfaces. The criterion *i_diff_* ≤ *i_dis,max_* for pit growth [30] is more likely to be satisfied and sustained in pit sites on rougher surfaces. As such, the metastable pit on a rougher surface would have a longer lifetime and larger pit size based on the metastable covered-pit model [30]. Besides, the larger difference between each individual pit site on a rougher surface makes the metastable pit radius distribution wider. Figure 17 describes schematically these aforementioned relationships between surface roughness, *i_diff_*, *i_dis,max_* and lifetime of a metastable pit for a fixed applied potential and temperature. However, the repassivation of a metastable pit is generally considered to be controlled by the rupture of the cover over the pit mouth [39,40] and may not occur as ideally as shown in Figure 17. Therefore, the distribution of final metastable pit sizes is less affected by the original pit morphology. As a result, the difference in metastable pit radius on varied surface roughness specimens is small compared to the former measured parameters, especially at lower temperatures.

#### 3.2.4. Influence of Surface Roughness on Metastable Pit Position

After the potentiostatic measurements, the position of the metastable pits was investigated. The BSE images of metastable pit position on the surface of different surface roughness samples are shown in Figure 18. The images illustrate that the metastable pits are initiated mainly in the ferrite phase, which appears as the darker phase for different surface roughness specimens. This result is mainly attributed to the variation of the pitting resistance equivalent number (PREN) of the ferrite and austenite phases in DSS 2205, which reveals the pitting corrosion resistance of alloys in solution containing chlorides [17,41]. The PREN is defined as:(9)PREN=wt % Cr+3.3 wt % Mo+16 wt % N

The preferential metastable pitting occurrence in the ferrite phase can be ascribed to the lower PREN value in this phase [29,41]; the difference in alloying element contents in the duplex phase leads to the relatively poor pitting corrosion resistance in the ferrite phase. However, comparison between different roughness specimens demonstrates that changing the macroscopic surface roughness, such as by mechanical grinding and polishing, is a physical factor that affects both ferrite and austenite phases of DSS uniformly and cannot influence the pitting location preference. In summary, the influence of surface roughness is mainly on the possibility of pit nucleation and pit growth dynamics, rather than pit initiation position.

## 4. Conclusions

In the present work, the influence of surface roughness on pitting corrosion resistance of DSS 2205 was investigated by EN techniques and statistical analysis. The metastable/stable pit initiation and growth on DSS are obviously affected by surface condition and were similar to those on austenite stainless steel. The specific conclusions obtained were as follows:Rougher surface conditions decreased the EN resistance *R_n_* of DSS 2205 and consequently increasesd the probability of stable pit occurrence.Based on the frequency domain analysis of EN, the pit nucleation was more frequent and the difference in pit initiation morphology was much larger on rougher-surfaced DSS 2205. Besides, the short noise analysis of the EN record showed that the pit growth process was more stable and the probability of metastable-to-stable transition was higher on rougher surfaces. However, the resolution of EN measurement was not sufficient to distinguish clearly the differences between specimens of different surface roughness.Potentiostatic EN measurement represents more obviously the effect of surface roughness on pitting corrosion of DSS. On rougher surfaces, the frequency of metastable pit initiation increases because pit sites on a rough surface are more likely to reach *C_cirit_*. The dispersity of the metastable pitting rate is also greater due to the more significant differences in initial pit sites on a rough surface.According to potentiosatic EN measurement, rougher surface increasd the growth rate of metastable pit propagation on DSS, as well as its SD. A model representing the relationship between effective diffusion length of a pit site and the diffusion rate of a metastable pit has been proposed to explain the variation of growth rate with surface roughness.The increase in surface roughness also enlarged the final size of a metastable pit on DSS. The SD of this parameter increased likewise. The final size of the metastable pit is attributed to the decrease in diffusion current density associated with the surface roughness.The variation of surface roughness did not affect the pit nucleation site preference on DSS 2205. The effects of surface roughness on the both phases of DSS are identical, and the pit location was determined mainly by other factors, such as the chemical content of the phases.

## Figures and Tables

**Figure 1 materials-12-00738-f001:**
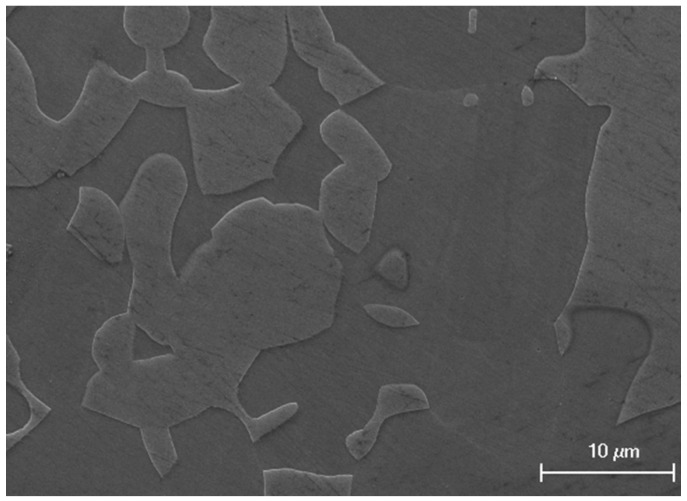
Microstructure of DSS 2205, solution-annealed at 1050 °C for 1 h, showing ferrite (dark) and austenite (bright) phases.

**Figure 2 materials-12-00738-f002:**
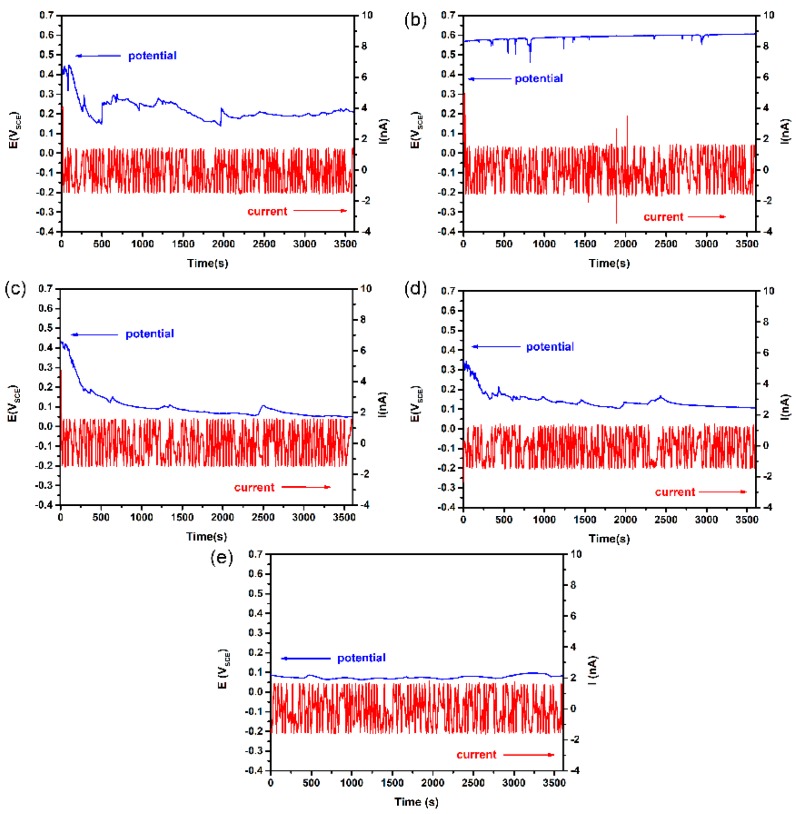
Typical EN patterns of DSS 2205 of different surface roughness conditions in 10% FeCl_3_ solution at 50 °C: (**a**) 180#; (**b**) 600#; (**c**) 1000#; (**d**) 2000#; € polished.

**Figure 3 materials-12-00738-f003:**
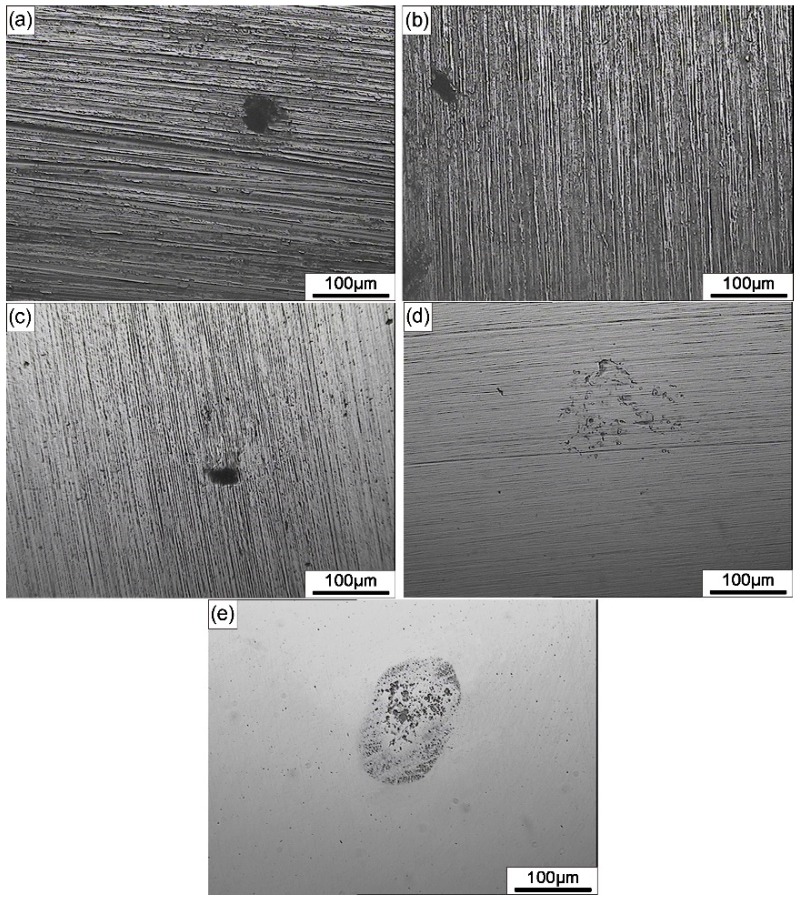
Optical microscopy images of the pits on different surface roughness specimens after EN measurement in 10% FeCl_3_ solution at 50 °C: (**a**) 180#; (**b**) 600#; (**c**) 1000#; (**d**) 2000#; (**e**) polished.

**Figure 4 materials-12-00738-f004:**
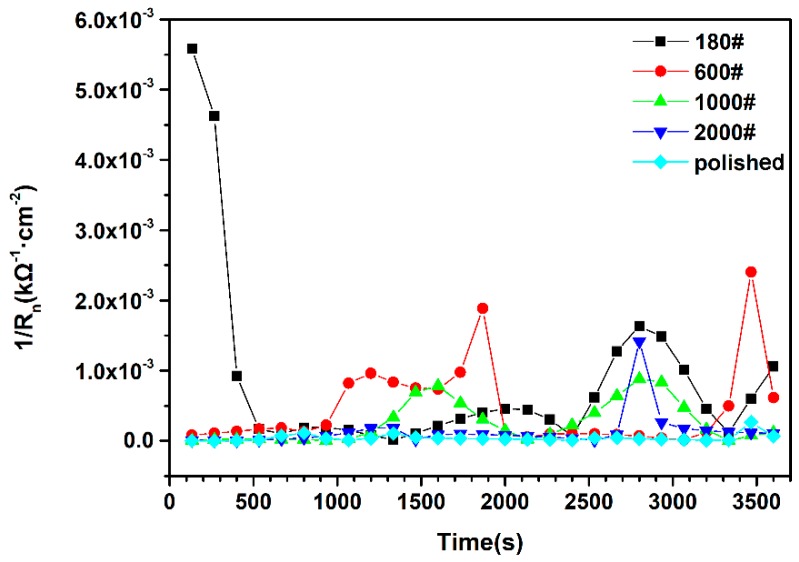
Reciprocal of *R_n_* of different surface roughness DSS 2205 samples in 10% FeCl_3_ at 50 °C.

**Figure 5 materials-12-00738-f005:**
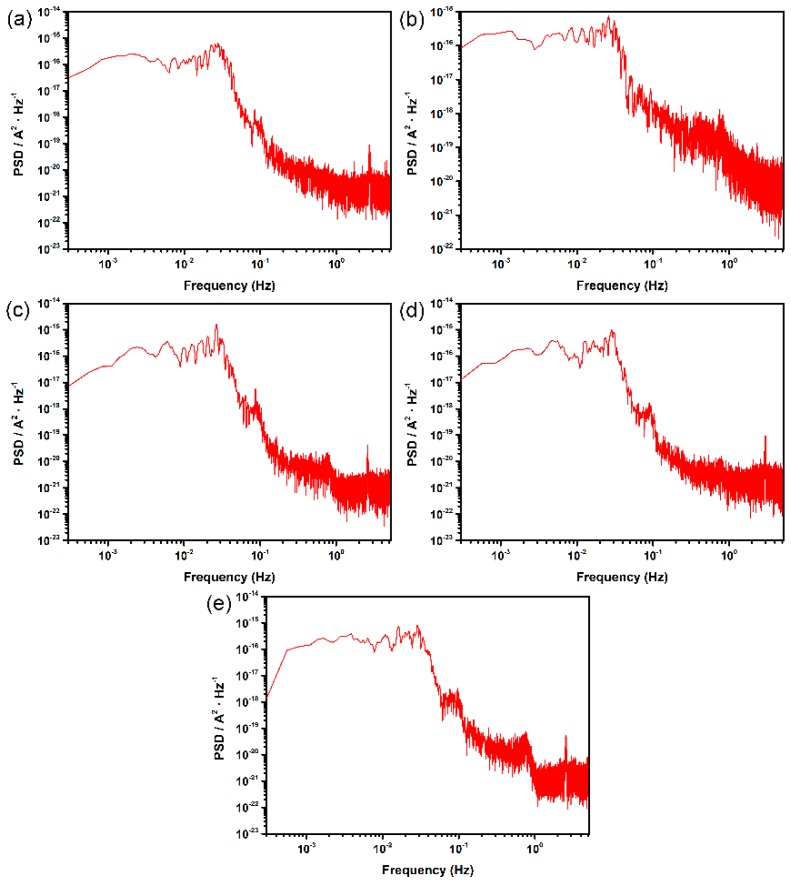
Current PSD plots for different surface roughness samples of DSS 2205 in 10% FeCl_3_ at 50 °C: (**a**) 180#; (**b**) 600#; (**c**) 1000#; (**d**) 2000#; (**e**) polished.

**Figure 6 materials-12-00738-f006:**
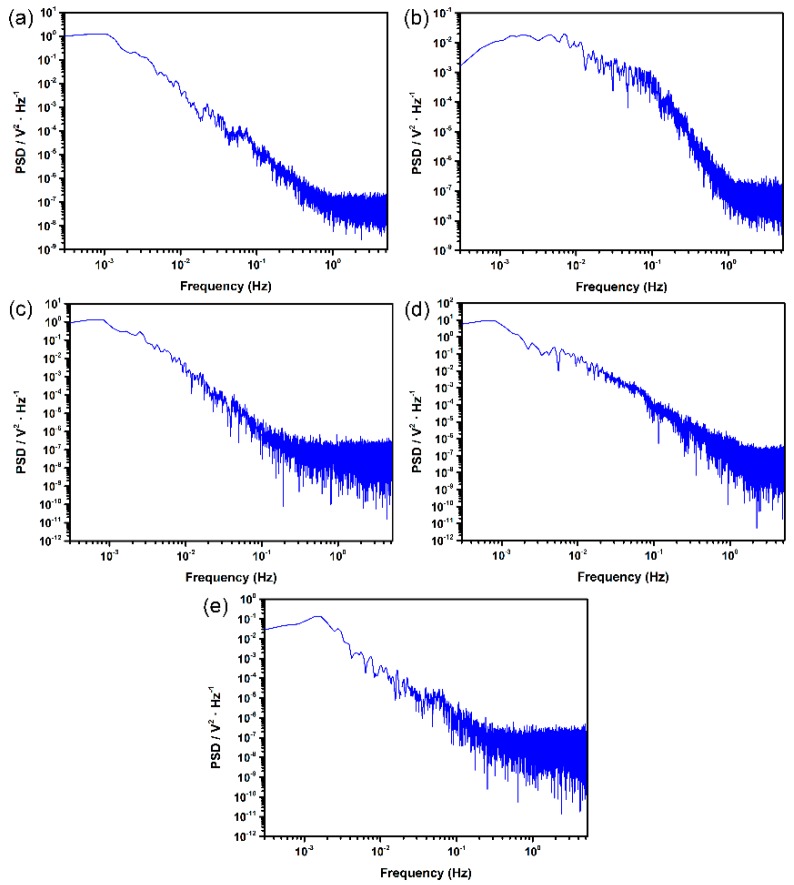
Potential PSD plots for different surface roughness samples of DSS 2205 in 10% FeCl_3_ at 50 °C: (**a**) 180#; (**b**) 600#; (**c**) 1000#; (**d**) 2000#; (**e**) polished.

**Figure 7 materials-12-00738-f007:**
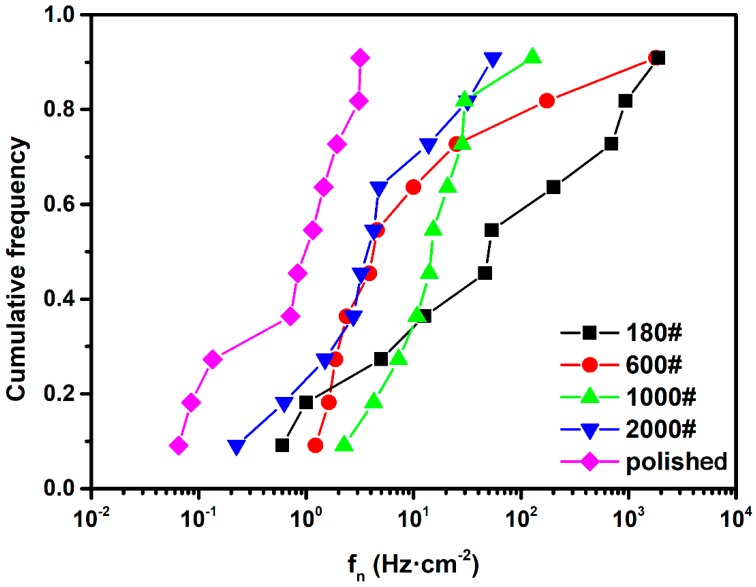
Cumulative frequency of *f_n_* for different surface roughness DSS 2205 specimens in 10% FeCl_3_ at 50 °C.

**Figure 8 materials-12-00738-f008:**
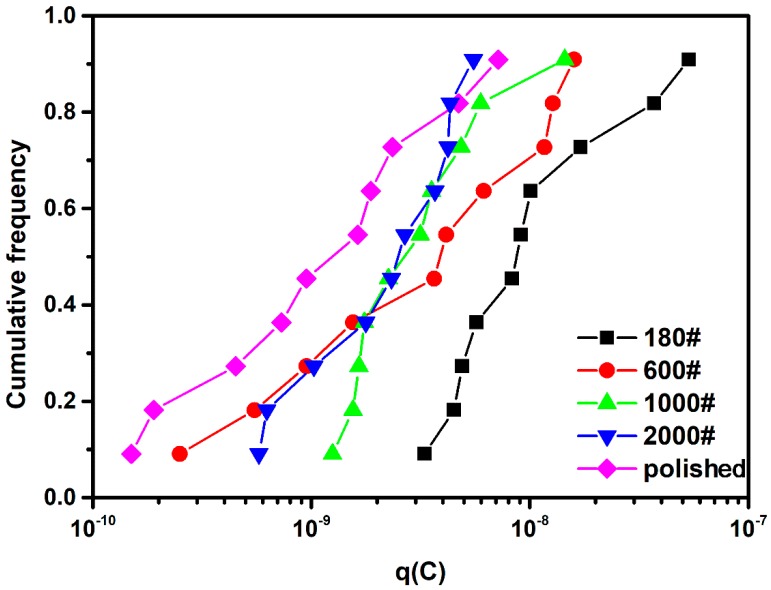
Cumulative frequency of *q* for different surface roughness DSS 2205 specimens in 10% FeCl_3_ at 50 °C.

**Figure 9 materials-12-00738-f009:**
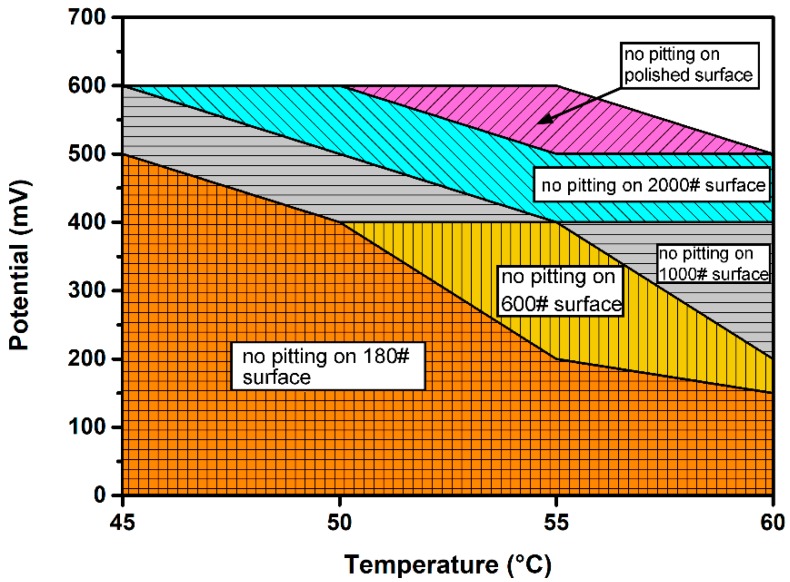
Potentiostatic measurements on different surface roughness specimens at various potentials and solution temperatures.

**Figure 10 materials-12-00738-f010:**
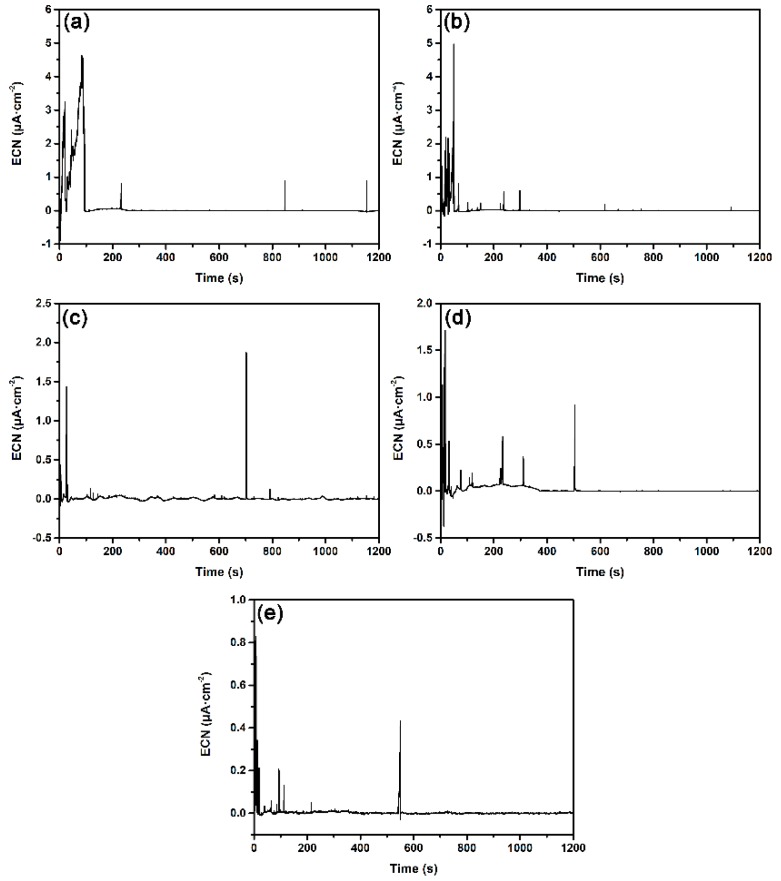
Typical EN time records of different surface roughness specimens of DSS 2205 at 200 mV (SCE) in NaCl solution at 55 °C: (**a**) 180#; (**b**) 600#; (**c**) 1000#; (**d**) 2000#; (**e**) polished.

**Figure 11 materials-12-00738-f011:**
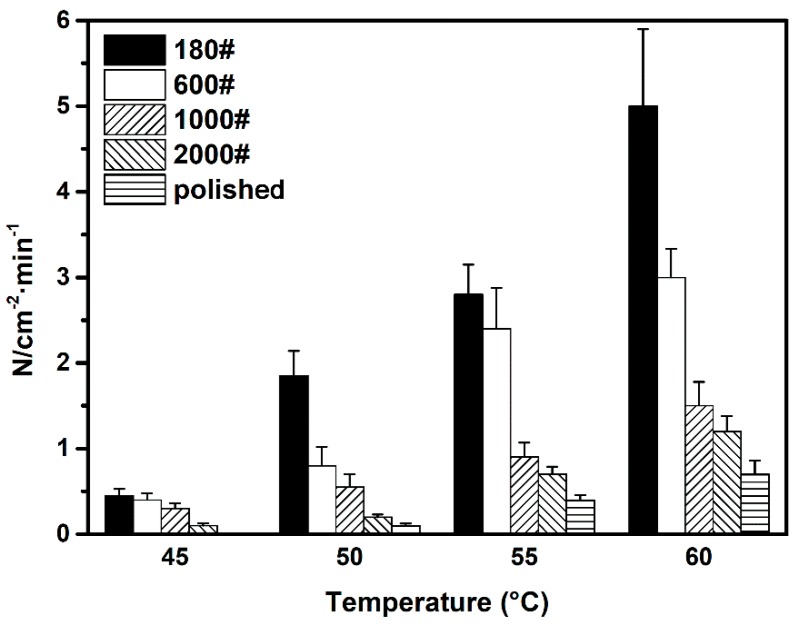
Metastable pitting rate of different surface roughness specimens measured at four testing temperatures.

**Figure 12 materials-12-00738-f012:**
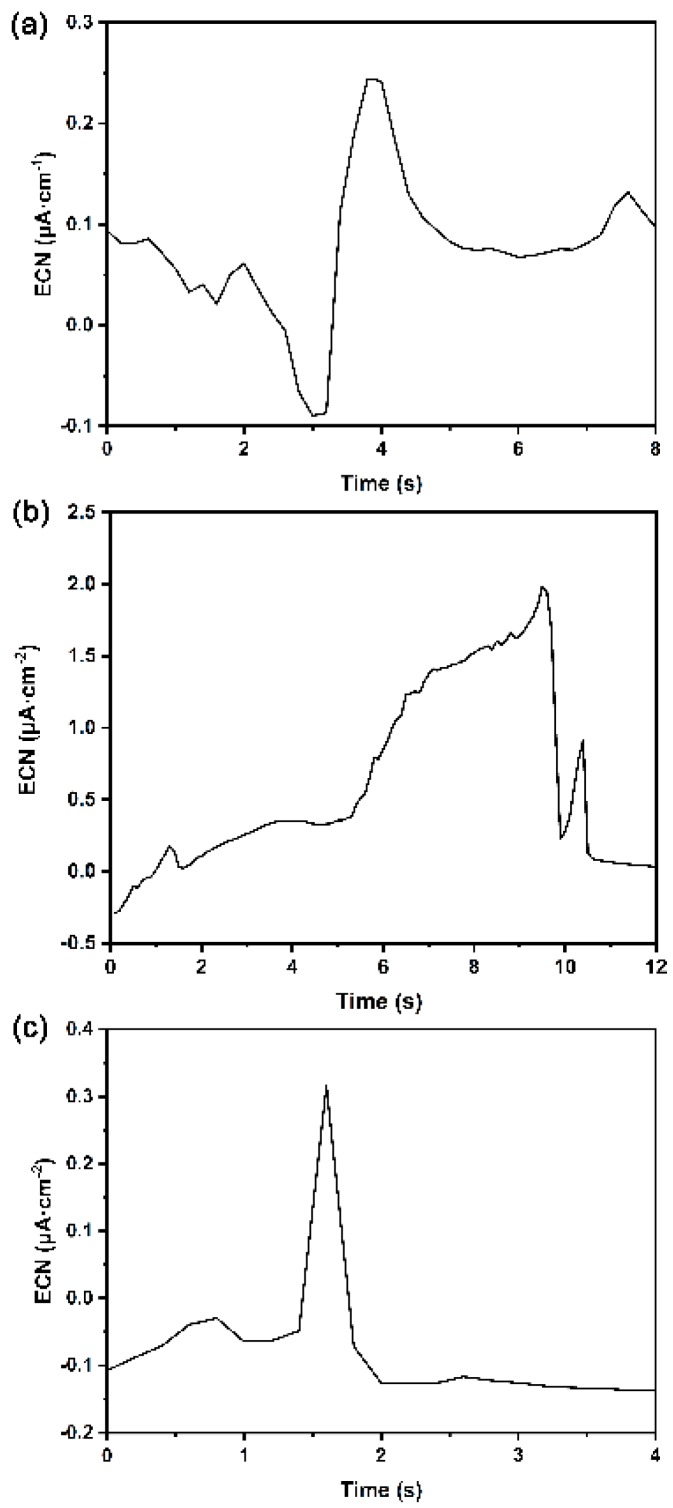
Different types of current transients observed on DSS2205 specimens in 1 M NaCl solution: (**a**) type I—sharp increase followed by slow decrease; (**b**) type II—slow increase followed by sharp decrease; (**c**) type III—symmetrical increase and decrease.

**Figure 13 materials-12-00738-f013:**
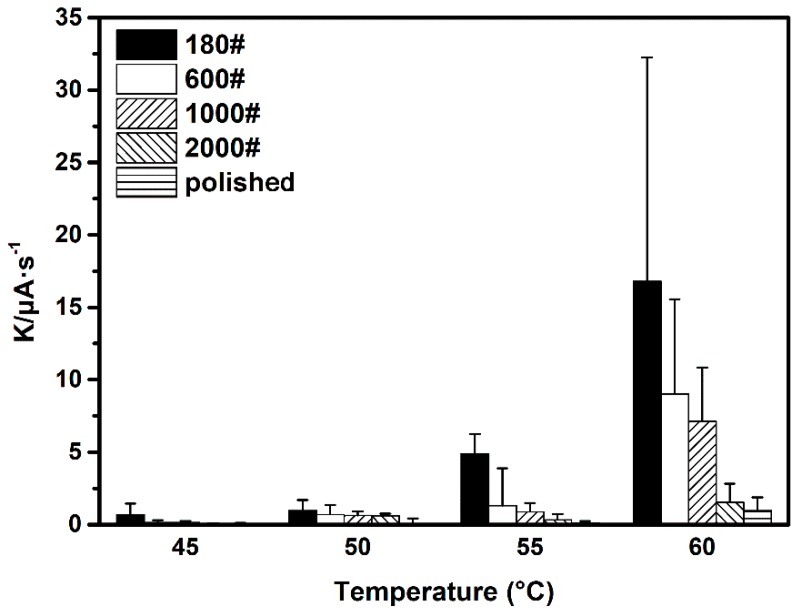
Metastable pit growth rate of different surface roughness DSS 2205 specimens at four testing temperatures.

**Figure 14 materials-12-00738-f014:**
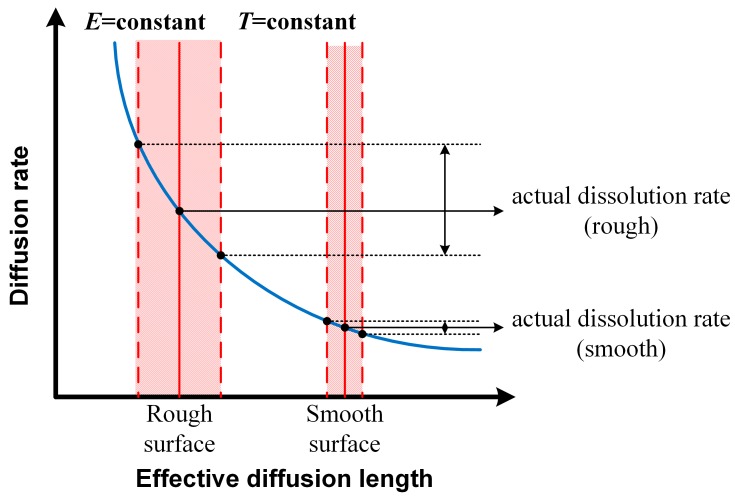
Schematic presentation of the relationships between surface roughness, effective diffusion length, diffusion rate, and actual dissolution rate of metastable pits.

**Figure 15 materials-12-00738-f015:**
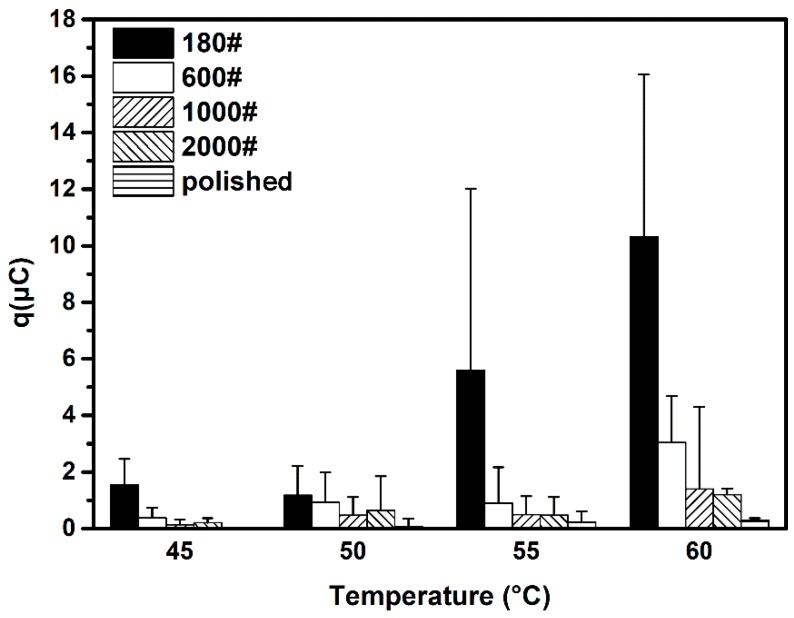
Average charge quantity of ECN transient of different surface roughness DSS 2205 specimens at four test temperatures.

**Figure 16 materials-12-00738-f016:**
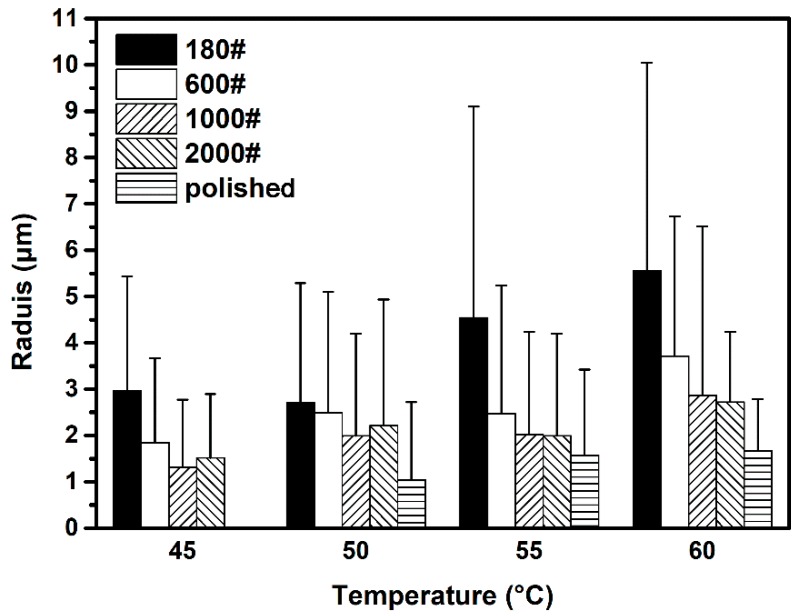
Average radius of metastable pit on different surface roughness DSS 2205 specimens at four solution temperatures.

**Figure 17 materials-12-00738-f017:**
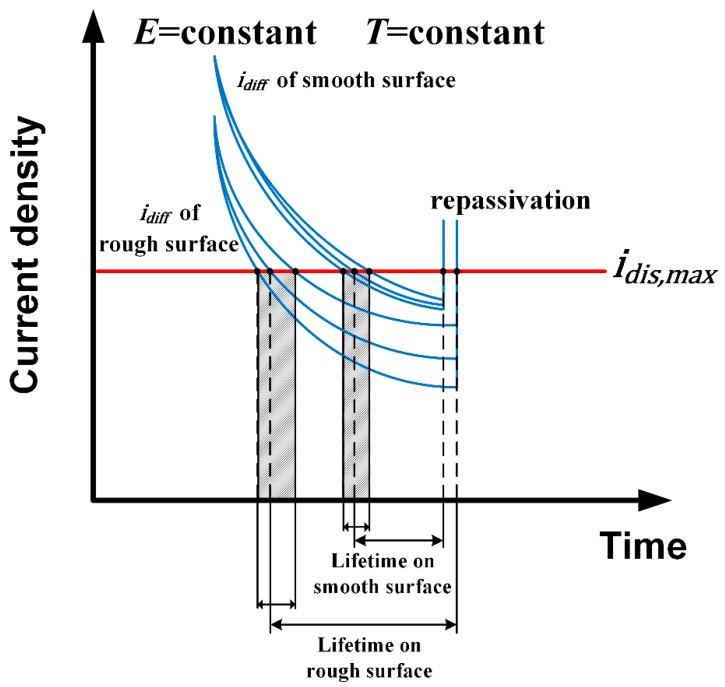
Schematic presentation of the relationships between surface roughness, idiff, idis,max, and lifetime of a metastable pit.

**Figure 18 materials-12-00738-f018:**
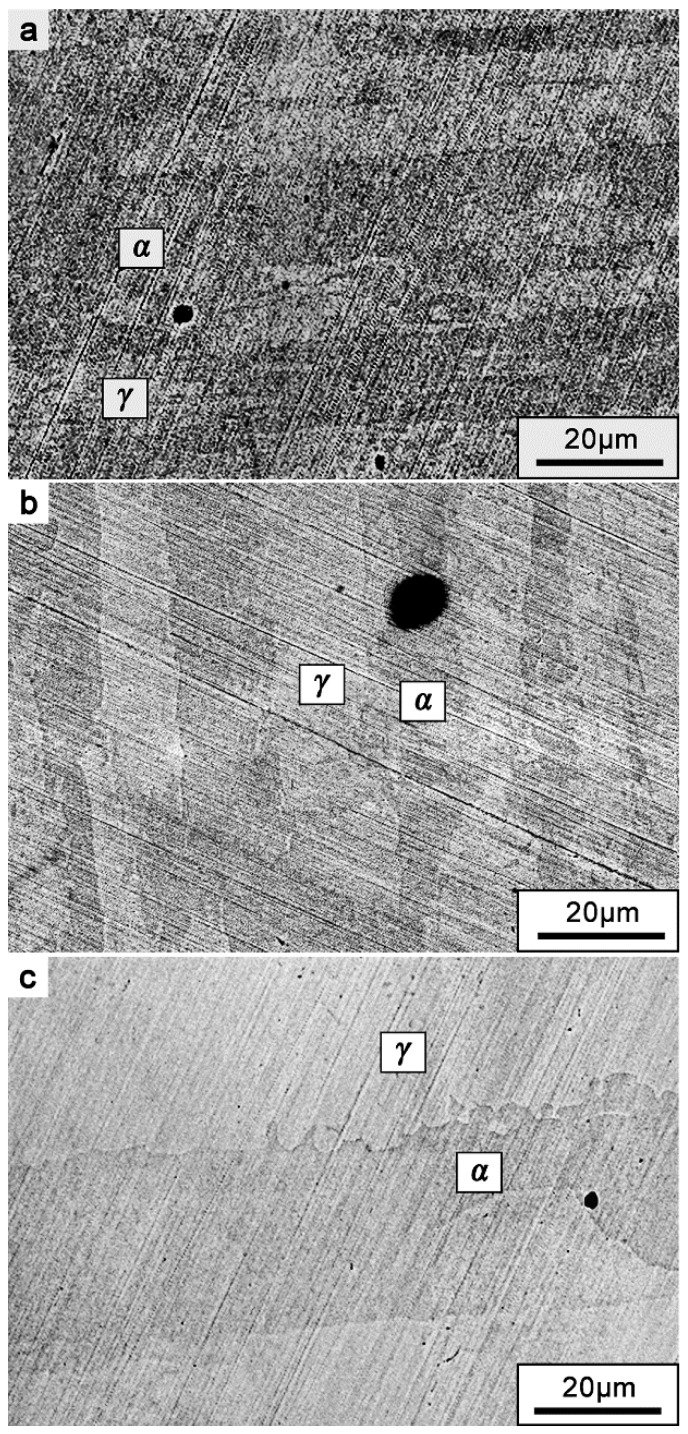
BSE image of metastable pit position on different surface roughness DSS 2205 specimens: (**a**) 180#; (**b**) 1000#; (**c**) polished.

**Table 1 materials-12-00738-t001:** The chemical composition (wt %) of DSS 2205.

Element	C	Si	Mn	P	S	Cr	Mo	Ni	N	Cu	Fe
Wt %	0.026	0.047	1.37	0.023	0.001	22.27	3.10	5.46	0.15	0.15	Bal.

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
