# Peer review of "Effect of Surface Roughness on Pitting Corrosion of 2205 Duplex Stainless Steel Investigated by Electrochemical Noise Measurements"

_materials, 2019, doi:10.3390/ma12050738_

Round 1

Reviewer 1 Report

This paper studies the effect of surface roughness on pitting corrosion in DSS 2205. The authors use fairly well established techniques to study this material. Overall they did a lot of work but the story is not very strong or clear. I suggest using about half of the figures and providing more in depth discussion about the results shown. The message really gets lost in all the figures and it would be better to make the paper more clear and concise. Critically assess what information is needed to support your conclusions. If you only 4 conclusions you probably only need 4-8 figures to support them, certainly not 17. 

A few specific suggestions:

Put plots on the same scale axis when possible to help the reviewer compare data.

The 3-D plots in figure 10, 14, 15 look nice but are rather useless for reading numbers or comparing data points. Consider another plotting style.

The authors mention that replicate tests were performed on the EN measurements but they are not shown or discussed. 

Avoid the acronym for austenitic stainless steels. ASS mean butt in slang english. 

Extensive review of the English and grammar are needed.

Critical pitting potential is given the acronym CPT in the introduction. It is used later for critical pitting temperature. Please correct.

Author Response

Response to Reviewer 1 Comments

Point 1: I suggest using about half of the figures and providing more in depth discussion about the results shown. The message really gets lost in all the figures and it would be better to make the paper clearer and more concise. Critically assess what information is needed to support your conclusions. If you only 4 conclusions you probably only need 4-8 figures to support them, certainly not 17.

Response 1: Thanks for your constructive suggestion. Just like what the reviewer said, the paper is better to be clear and concise with less figures and more in-depth discussion. But we used 17 figures in order to exhibit the research process and results completely. Figure 1 and 2 displayed the results of EN measurements at OCP. Figure 3 showed the time-domain analysis of EN measurements. Figure 4 to 7 represented the time-frequency transform and frequency-domain analysis related to initiation and growth of pitting corrosion. Figure 8 exhibit the optimization of potentiostatic EN measurement. Figure 9 to 17 included the analysis of potentiostatic EN results, the discussion of the influence on surface roughness on metastable pit nucleation, growth and pit location in the dual phase structure. The potentiostatic EN measurements were the supplement of EN at OCP and further investigation of the mechanism. According to reviewer’s advice, we have deleted some useless descriptions and rewritten the conclusions to make the paper clearer and all figures meaningful. Thank you for your kind and instructive comment again.

Point 2: Put plots on the same scale axis when possible to help the reviewer compare data.

Response 2: Thanks for your helpful comment. We have changed axis in the plot to the same scale.

Point 3: The 3-D plots in figure 10, 14, 15 look nice but are rather useless for reading numbers or comparing data points. Consider another plotting style.

Response 3: Thanks for your valuable advice. We have changed Figure 10, 14, 15 into 2-D plots to increase the legibility.

Point 4: The authors mention that replicate tests were performed on the EN measurements, but they are not shown or discussed.

Response 4: Thank you for the kind evaluation. The replicate tests of EN measurement at OCP were to ensure the results were not interfered by external noise signals. Because the EN analysis in essence is a statistical method of EN signal, the time-domain and frequency-domain analysis of measured EN data in this work has already included the results from parallel samples. The replicate tests of potentiostatic EN measurements were represented in the error bar of the measured parameters in Figure 10, 12, 14 and 15. The meaning of it in pitting corrosion behaviour was also discussed. Thanks again for your comment, we have rewritten some parts to make the description clearer.

Point 5: Avoid the acronym for austenitic stainless steels. ASS mean butt in slang english.

Response 5: We have made correction according to the Reviewer’s comment. The austenite stainless steel in the paper has been modified to full spelling instead of the improper acronym.

Point 6: Extensive review of the English and grammar are needed.

Response 6: Thanks very much. According to your comment, we have polished the manuscript.

Point 7: Critical pitting potential is given the acronym CPT in the introduction. It is used later for critical pitting temperature. Please correct.

Response 7:  We are sorry for the incorrect writing in the introduction. The acronym CPT stands for critical pitting temperature. We have corrected this mistake.

Reviewer 2 Report

The article analyzes in depth the effect of the surface finishing on the pitting corrosion behavior of the DSS 2205 alloy, using the Electrochemical Noise Measurements technique. The article can be suitable for the publication, even if it does not present any significant new information from the scientific point of view, because this type of analysis has already been made for austenitic steels, as also highlighted by the authors. The authors have discussed that the localized corrosive attack mainly started from the ferrite phase, independently from the surface roughness, because of its lower resistance to pitting corrosion (lower PREN value) compared to the austenite phase. Maybe in the analysis presented by the authors, a higher emphasis should be given to this observation, with some additional data, taking into consideration that the biphasic microstructure characterizes this alloy. Some further advices follow for improving manuscript before the publication.

- English editing is necessary for improving the manuscript readability. Some sentences are not very fluent, for example line 20-21 in the introduction paragraph, and some grammar errors are present thorough all the text.

- Table 1 is presented twice in the Material and Specimen Preparation section.

-I suggest to use the same range of values for the potential (axis of the ordinates) in the graphs of Figure 1, to better highlight the differences in the potential values recorded for the different levels of roughness.

- Add some references to sustain the CPT data indicated in line 217-218.

-In my opinion, it is difficult to see that the pit started in ferrite phase from Figure 17-a. Please select a clearer image.

Author Response

Response to Reviewer 1 Comments

Point 1: The authors have discussed that the localized corrosive attack mainly started from the ferrite phase, independently from the surface roughness, because of its lower resistance to pitting corrosion (lower PREN value) compared to the austenite phase. Maybe in the analysis presented by the authors, a higher emphasis should be given to this observation, with some additional data, taking into consideration that the biphasic microstructure characterizes this alloy. Some further advices follow for improving manuscript before the publication.

Response 1: Thanks for your constructive suggestion. As the reviewer said, the effect of dual phase structure is a research priority in pitting corrosion of duplex stainless steel (DSS), and some researcher has already focused on this issues. This work slightly involves this, but the main purpose of this paper is to investigate the influence of surface roughness on the dynamic process of pitting initiation and growth on DSS. Nevertheless, influence of phase structure on the pitting corrosion behaviour under different surface conditions can be further investigate in our following research. Thanks for your good comment again.

Point 2: English editing is necessary for improving the manuscript readability. Some sentences are not very fluent, for example line 20-21 in the introduction paragraph, and some grammar errors are present thorough all the text.

Response 2: Thanks for your helpful comment. We have polished the manuscript and corrected the mistakes.

Point 3: Table 1 is presented twice in the Material and Specimen Preparation section.

Response 3: Thanks for your valuable advice. We have changed Figure 10, 14, 15 into 2-D plots to increase the legibility.

Point 4: I suggest using the same range of values for the potential (axis of the ordinates) in the graphs of Figure 1, to better highlight the differences in the potential values recorded for the different levels of roughness.

Response 4: Thank you for the helpful suggestion. We have changed the ordinate axis in Figure 1 to the same scale.

Point 5: Add some references to sustain the CPT data indicated in line 217-218.

Response 5: Thanks for your instructive advice. We have added related references to support the CPT data.

Point 6: In my opinion, it is difficult to see that the pit started in ferrite phase from Figure 17-a. Please select a clearer image.

Response 6: Thanks to point the image quality issue. Due to the rough surface of the specimen, the phase structure is hard to be characterized with good resolution. We have tried other characterization methods, but the effect was unsatisfied. The figure we submitted was the clearest one. We hope you can understand.

Reviewer 3 Report

1. The role of surface roughness in stable and metastable pitting corrosion behavior has been widely investigated. In this paper, the authors stated in Introduction as follows; 'it is doubted whether the conclusion can be fully extended to duplex stainless steel.'. But, only the predictable results were drawn in this paper and the unique corrosion behavior of dual-phased steel was not discussed. 2. Revise Table 1. Present precise and measured chemical composition of the investigated alloy, instead of the composition range. 3. Present the phase diagram of the investigated alloy, and explain the temperature for solution annealing (that is, 1050oC, in this paper). As far as I know, the appropriate temperature for the solution annealing and controlling the phase fraction for UNS S32205 alloy is higher than 1050 degree. 4. The microstructure of the solution annealed specimen should be presented. 5. In the experimental section, ZRA (WE-WE-RE) in FeCl3 and potentiostatic test (WE-CE-RE) in 1 M NaCl solution were described. Where are the results from both tests? 6. The critical pitting temperature (ASTM G48) of UNS S32205 is less than 50 oC. Thus, explain why the authors chose the extremely corrosive environment (10% FeCl3-6H2O, 50oC) for this work. 7. The test condition for Figure 1 should be explained in caption. In addition, explain the relationship between the potential-time curve and current-time curve and the physical meaning of the highly-noisy current-time curve.

Author Response

Response to Reviewer 3 Comments

Point 1: In this paper, the authors stated in Introduction as follows; 'it is doubted whether the conclusion can be fully extended to duplex stainless steel.'. But only the predictable results were drawn in this paper and the unique corrosion behavior of dual-phased steel was not discussed.

Response 1: Thank you for your constructive criticism. This work focused on the influence of surface roughness on the pitting corrosion behaviour of duplex stainless steel (DSS), which is seldom reported by other researchers. Different surface roughness was performed by mechanical polishing to simulate practical process, such as shot peening. Before the experimental measurements, we were not sure whether the relation between surface roughness and pitting corrosion resistance on DSS is same as that on austenite stainless steel or not. Due to the dual phase structure and the effect of relatively weak phase, we guessed that pitting corrosion resistance of DSS might have the minimum/maximum value or remain unchanged at first as surface roughness increased. But after the electrochemical measurements and noise analysis, the trend of DSS was consistent with austenite stainless steel. The pit position characterization also indicated that the mechanical polishing did not change the phase structure of DSS, the two phases were influenced by surface roughness equally. Hence the unique corrosion behaviour of DSS was not presented. We are sorry for not expressing the point in the paper. According to reviewer’s thoughtful comment, we have added the related description in the discussion and conclusion section.

Point 2: Revise Table 1. Present precise and measured chemical composition of the investigated alloy, instead of the composition range.

Response 2: We have revised Table 1 according to the reviewer’s comment and provided measured composition of 2205 duplex stainless steel.

Point 3: Present the phase diagram of the investigated alloy, and explain the temperature for solution annealing (that is, 1050 oC, in this paper). As far as I know, the appropriate temperature for the solution annealing and controlling the phase fraction for UNS S32205 alloy is higher than 1050 degree.

Response 3: Thanks for your valuable suggestion. The solution annealing temperature chosen for DSS 2205 was based on other researcher’s published paper. Solution annealing at temperature above 1050 oC would lead to the precipitation of Cr and N rich phases at ferrite matrix and deteriorate the pitting resistance of DSS 2205, thus 1050 oC is the relatively suitable annealing temperature for DSS 2205. Many researchers also have used this temperature for annealing in pre-treatment of DSS2205. We have added the explanation of this temperature and cited the related references in experimental section according to your helpful advice. Because this manuscript did not involve the discussion of heat treatment and precipitated phase of duplex stainless steel, the phase diagram was not presented considering the space limitation. We think the cited references and description can explain the reason of choosing the 1050 oC, hope you will understand.

Point 4: The microstructure of the solution annealed specimen should be presented.

Response 4: Thank you for your instructive suggestion. We have added the microstructure of the solution annealed specimen in the “Materials and Methods” section.

Point 5: In the experimental section, ZRA (WE-WE-RE) in FeCl3 and potentiostatic test (WE-CE-RE) in 1 M NaCl solution were described. Where are the results from both tests?

Response 5: Thanks for your thoughtful comment. ZRA in FeCl3 was used to record the electrochemical noise at open circuit potential, the original record is displayed in Figure 1 and the analysis of ZRA result is presented in section 3.1, including the time-domain and frequency-domain analysis of electrochemical noise at OCP. The original results of potentiostatic EN test is exhibited in Figure 9, and the current noise signals were analysed in section 3.2 to investigate the influence of surface roughness on metastable nucleation and propagation of 2205 duplex stainless steel.

Point 6: The critical pitting temperature (ASTM G48) of UNS S32205 is less than 50 oC. Thus, explain why the authors chose the extremely corrosive environment (10% FeCl3-6H2O, 50 oC) for this work.

Response 6: Thanks for your constructive comment.  The ASTM G48 method of UNS S32205 requires the test time of 24h, and the electrochemical noise (EN) measurement only last one hour, thus the relatively corrosive environment would not lead to serious pitting corrosion on EN specimens. The images of specimens after EN measurement also proved the applicability of this solution environment. The reason of choosing this testing environment was to ensure effective EN signal of pitting process can be obtained during recording. In addition, this environmental condition was used by other researchers to investigate the localized corrosion of duplex stainless steel. We have cited related reference in the description of experiment according to reviewer’s suggestion.

Point 7: The test condition for Figure 1 should be explained in caption. In addition, explain the relationship between the potential-time curve and current-time curve and the physical meaning of the highly-noisy current-time curve.

Response 7:  Thank you for your careful reading of the manuscript. According to reviewer’s comment, we have added the test condition in caption of Figure 1. We are also sorry for not describing the relationship between current noise and potential noise and the meaning of highly-noisy current-time curve, the related part has been rewritten.

Round 2

Reviewer 3 Report

The authors’ response and the modifications in this manuscript are not sufficient and inacceptable. The authors maybe take the several previous comments lightly. In this revised version, there is no phase diagram (space limitation cannot be a reason) and no discussion about the microstructures of duplex stainless steel. In addition, the inappropriate test methods are not explained clearly. More importantly, I cannot find any potential-current correlation in Figure 1! In the current-time curve, the minus current also should be explained. Still, I don’t think this paper provides new scientific discovery, but the authors insist that the originality of this paper is in the ‘duplex’ phased materials. If so, the microstructure of the investigated alloy should be more carefully defined and discussed at least. Moreover, based on the response letter, I have to tell that the interpretation of the corrosion phenomena in this paper is wrong. Thus I have to reject this paper for publication.

Author Response

Thanks for your report.